# Thermodynamic, Physical, and Structural Characteristics in Layered Hybrid Type (C_2_H_5_NH_3_)_2_*M*Cl_4_ (*M* = ^59^Co, ^63^Cu, ^65^Zn, and ^113^Cd) Crystals

**DOI:** 10.3390/molecules25081812

**Published:** 2020-04-15

**Authors:** Ae Ran Lim

**Affiliations:** Analytical Laboratory of Advanced Ferroelectric Crystals and Department of Science Education, Jeonju University, Jeonju 55069, Korea; aeranlim@hanmail.net or arlim@jj.ac.kr; Tel.: +82-(0)63-220-2514

**Keywords:** crystal growth, organic-inorganic hybrid compound, (C_2_H_5_NH_3_)_2_CoCl_4_, thermodynamic, MAS NMR

## Abstract

The thermal, physical, and molecular dynamics of layered hybrid type (C_2_H_5_NH_3_)_2_*M*Cl_4_ (*M* = ^59^Co, ^63^Cu, ^65^Zn, and ^113^Cd) crystals were investigated by thermogravimetric analysis (TGA) and magic angle spinning nuclear magnetic resonance (MAS NMR) spectroscopy. The temperatures of the onset of partial thermal decomposition were found to depend on the identity of *M*. In addition, the Bloembergen–Purcell–Pound curves for the ^1^H spin-lattice relaxation time T_1ρ_ in the rotating frames of CH_3_CH_2_ and NH_3_, and for the ^13^C T_1ρ_ of CH_3_ and CH_2_ were shown to exhibit minima as a function of the inverse temperature. These results confirmed the rotational motion of ^1^H and ^13^C in the C_2_H_5_NH_3_ cation. Finally, the T_1ρ_ values and activation energies E_a_ obtained from the ^1^H measurements for the H‒Cl···*M* (*M* = Zn and Cd) bond in the absence of paramagnetic ions were larger than those obtained for the H‒Cl···*M* (*M* = Co and Cu) bond in the presence of paramagnetic ions. Moreover, the E_a_ value for ^13^C, which is distant from the *M* ions, was found to decrease upon increasing the mass of the *M* ion, unlike in the case of the E_a_ values for ^1^H.

## 1. Introduction

Layered hybrid compounds have drawn great attention as a new generation of high performance materials due to their interesting physical and chemical properties obtained through the combination of organic and inorganic materials at the molecular level [1,2,3]. They consist of a wide range of inorganic anion chains, alternating with a large variety of organic cations as building blocks. The organic component of the hybrid complex provides several useful properties, such as structural flexibility and optical properties, while the inorganic part is responsible for the mechanical and thermal stabilities, in addition to interesting magnetic and dielectric transitions [4,5]. The diversity of such hybrid materials is therefore large, and so offers a wide range of structures, properties, and potential applications [6,7,8,9,10,11]. More specifically, hybrid layered compounds based on the perovskite structure are interesting materials due to their potential application in solar cells [2,3]. However, the toxicity and chemical instability of halide perovskites limit their use. As a result, the replacement of the lead in present in the perovskite structure with alternative cost-effective materials that are environmentally friendly, less-toxic, and more readily available (e.g., transition metals) is necessary for the extended application of perovskites in solar cells [3]. The structure of (C*_n_*H_2*n*+1_NH_3_)_2_*M*Cl_4_ compounds, where *n* = 1, 2, 3… and *M* represents a divalent metal (*M* = Co^2+^, Cu^2+^, Zn^2+^, and Cd^2+^), has been described as a sequence of alternating organic-inorganic layers [2,3,12]. The structures of (C_2_H_5_NH_3_)_2_*M*Cl_4_ crystals with *n* = 2 are similar within each group but dissimilar between groups due to differences between either the inorganic or organic components. For example, the inorganic frames where *M* = Cu^2+^ and Cd^2+^ are corner-sharing *M*Cl_6_ octahedra, while those of *M* = Co^2+^ and Zn^2+^ are simple *M*Cl_4_ tetrahedra [13]. In addition, the organic chains are joined by weak hydrogen bonds between the NH_3_ groups and the Cl ions. Indeed, the structural geometries and molecular dynamics of the organic molecules within the layered hybrid structures are important for determining the influence of temperature on the structural phase transitions.

As an example, (C_2_H_5_NH_3_)_2_CoCl_4_ crystallizes as an orthorhombic structure, which undergoes a reversible phase transition at 235 K [14]. In addition, (C_2_H_5_NH_3_)_2_CuCl_4_ undergoes phase transitions at 236, 330, 357, and 371 K [7,15,16,17,18,19], its crystal structure at room temperature is orthorhombic [20]. In contrast, (C_2_H_5_NH_3_)_2_ZnCl_4_ undergoes five phase transitions at 231, 234, 237, 247, and 312 K [21], crystallizing as an orthorhombic system at room temperature [22]. Finally, (C_2_H_5_NH_3_)_2_CdCl_4_ undergoes structural phase transitions at 114, 216, 358, and 470 K [9,23,24], whereby the room temperature orthorhombic phase has the *Abma* space group [23]. The structure of the organic component consists of a double layer of alkylammonium ions with the charged nitrogen atoms oriented to the nearest *M*Cl_4_ tetrahedra or *M*Cl_6_ octahedra. The phase transition temperatures, lattice constants, structures, and space groups for the four crystals are summarized in Table 1.

Based on our previously reported nuclear magnetic resonance (NMR) results, the molecular dynamics of the cation present in (C_2_H_5_NH_3_)_2_*M*Cl_4_ (*M* = Cu, Zn, and Cd) crystals were discussed in terms of temperature-dependent chemical shifts and spin-lattice relaxation times T_1ρ_ in the rotating frames for the ^1^H and ^13^C nuclei [25,26,27].

Thus, to better elucidate the thermal stability in (C_2_H_5_NH_3_)_2_CoCl_4_ single crystals grown by the slow evaporation method, we herein describe the use of thermogravimetric analysis (TGA), in addition to structural analysis by variable-temperature ^1^H magic angle spinning (MAS) NMR spectroscopy and ^13^C cross-polarization (CP/MAS) NMR spectroscopy. Furthermore, the spin-lattice relaxation times T_1ρ_ in the rotating frames are measured for the ^1^H and ^13^C nuclei to better understand the physical and structural properties of (C_2_H_5_NH_3_)_2_CoCl_4_. The obtained results are compared with those of the previously reported (C_2_H_5_NH_3_)_2_CuCl_4_, (C_2_H_5_NH_3_)_2_ZnCl_4_, and (C_2_H_5_NH_3_)_2_CdCl_4_, and the properties dependent on the characteristics of the metal anion and the organic cation are identified.

## 2. Results and Discussion

### 2.1. Thermal Stability

The thermal stabilities of the various (C_2_H_5_NH_3_)_2_*M*Cl_4_ were examined by TGA, and the results are presented in Figure 1. Upon comparison of the TGA results with the possible chemical reactions taking place, the solid residues formed for (C_2_H_5_NH_3_)_2_*M*Cl_4_ were calculated based on Equations (1)–(4) [28]:
(C_2_H_5_NH_3_)_2_CoCl_4_ → (C_2_H_5_NH_2_)_2_CoCl_2_ (s) + 2HCl (g)Residue: (C_2_H_5_NH_2_)_2_CoCl_2_/(C_2_H_5_NH_3_)_2_CoCl_4_ = 74.03%(C_2_H_5_NH_3_)_2_CoCl_4_ → CoCl_2_ (s) + 2HCl (g) + 2(C_2_H_5_NH_2_) (g)Residue: CoCl_2_/(C_2_H_5_NH_3_)_2_CoCl_4_ = 46.24%(1)
(C_2_H_5_NH_3_)_2_CuCl_4_ → (C_2_H_5_NH_2_)_2_CuCl_2_ (s) + 2HCl (g)Residue: (C_2_H_5_NH_2_)_2_CuCl_2_/(C_2_H_5_NH_3_)_2_CuCl_4_ = 75.49%(C_2_H_5_NH_3_)_2_CuCl_4_ → CuCl_2_ (s) + 2HCl (g) + 2(C_2_H_5_NH_2_) (g)Residue: CuCl_2_/(C_2_H_5_NH_3_)_2_CuCl_4_ = 45.19%(2)
(C_2_H_5_NH_3_)_2_ZnCl_4_ → (C_2_H_5_NH_2_)_2_ZnCl_2_ (s) + 2HCl (g)Residue: (C_2_H_5_NH_2_)_2_ZnCl_2_/(C_2_H_5_NH_3_)_2_ZnCl_4_ = 75.64%(C_2_H_5_NH_3_)_2_ZnCl_4_ → ZnCl_2_ (s) + 2HCl (g) + 2(C_2_H_5_NH_2_) (g)Residue: ZnCl_2_/(C_2_H_5_NH_3_)_2_ZnCl_4_ = 45.53%(3)
(C_2_H_5_NH_3_)_2_CdCl_4_ → (C_2_H_5_NH_2_)_2_CdCl_2_ (s) + 2HCl (g)Residue: (C_2_H_5_NH_2_)_2_CdCl_2_/(C_2_H_5_NH_3_)_2_CdCl_4_ = 78.95%(C_2_H_5_NH_3_)_2_CdCl_4_ → CdCl_2_ (s) + 2HCl (g) + 2(C_2_H_5_NH_2_) (g)Residue: CdCl_2_/(C_2_H_5_NH_3_)_2_CdCl_4_ = 52.92%(4)


For the *M* = Co, Cu, Zn, and Cd species, the first mass losses were observed at approximately 378, 430, 460, and 550 K, respectively, which represent the onset of partial thermal decomposition, T_d_. From the results calculated using the molecular weights, mass losses of 25.97, 24.51, 24.36, and 21.05% for the different *M* ions were attributed to decomposition of the 2HCl moieties. These results are consistent with the TGA experiment results shown by dotted lines in Figure 1. Moreover, the final decomposition product is *M*Cl_2_, which corresponds to mass losses of 53.76, 54.81, 54.47, and 47.08%. These results indicate some differences between the calculated and experimental values. The difference between the calculation and experimental value of the final decomposition product is presumably dependent on the heating rate in the TGA experiment. Another difference is thought to be due to experimental conditions in air or N_2_ atmosphere. The decomposition temperature, T_d_, and mass loss of 2HCl, and final decomposition product for four crystals are summarized in Table 2.

Optical polarizing microscopy was used in order to determine whether these transformations are structural phase transitions or chemical reactions, as presented in Figure 2. In the case of (C_2_H_5_NH_3_)_2_CoCl_4_, the crystals are blue at room temperature, and no change in the crystal state was observed upon increasing temperature to 360 or 460 K, although melting was observed to commence at 465 K. In contrast, the (C_2_H_5_NH_3_)_2_CuCl_4_ crystals are dark yellow at room temperature, although they present a slightly inhomogeneous hue due to surface roughness. Upon increasing the temperature, the crystal color changed from dark yellow (300 K), to brown (380 K), to dark brown (450 and 500 K), and start melting was observed at 530 K. Interestingly, the crystals of (C_2_H_5_NH_3_)_2_ZnCl_4_ remained colorless and transparent (300, 450, and 460 K), and melting was observed between 470 and 475 K. Similarly, in the case of (C_2_H_5_NH_3_)_2_CdCl_4_, the crystals remained colorless and transparent between 300 and 480 K, although they became slightly opaque at approximately 540 K, prior to becoming fully opaque close to 570 K. Here, the sample temperatures shown in Figure 2 were kept constant during 2 min each temperature. For all four crystals, it was apparent that the phenomenon above T_d_ was not related to any structural phase transitions, but rather to a thermal decomposition, suggested by Lee [29].

### 2.2. Investigation of the Structural Properties and Molecular Dynamics by ^1^H MAS NMR

The ^1^H MAS NMR spectra of (C_2_H_5_NH_3_)_2_CoCl_4_ were recorded at a range of temperatures as shown in Figure 3. More specifically, at 300 and 370 K, the ^1^H signals for C_2_H_5_ and NH_3_ could not be distinguished, and the superimposed peak was rather broad; at 300 and 370 K, single peaks were observed at δ = 1.68 and δ = 0.02 ppm, respectively. In Figure 3, the spinning sidebands for the protons of C_2_H_5_NH_3_ are marked with asterisks. At 420 and 430 K, signals with chemical shifts of δ = 1.76 and 4.36 ppm, and δ = 1.79 and 4.37 ppm, were observed, respectively, which represent the protons of the C_2_H_5_ and NH_3_ ions. In addition, at these higher temperatures, the obtained signals became more intense, and the full-width at half-maximum (FWHM) values narrowed significantly, which were attributed to a high internal mobility.

The magnetization recovery traces for both the C_2_H_5_ and NH_3_ protons in (C_2_H_5_NH_3_)_2_CoCl_4_ can be described by a single exponential function [30,31]
P(*t*)/P_0_ = exp(‒*t*/T_1ρ_)(5)
where P(*t*) is the magnetization as a function of the spin-locking pulse duration *t*, and P_0_ is the total nuclear magnetization of the proton at thermal equilibrium. The recovery traces of the ^1^H nuclei for delay times ranging from 1 μs to 50 ms at 300 K are presented in the inset of Figure 4. Here, the asterisks represent spinning sidebands for the center peak. The T_1ρ_ values were obtained from the slopes of the delay time vs. the signal intensity, and were plotted as a function of the inverse temperature in Figure 4. As shown, the T_1ρ_ values sharply decrease close to 430 K, while near the phase-transition temperature T_C_, no changes are evident. At higher temperatures, the T_1ρ_ values for the C_2_H_5_ and NH_3_ protons were comparable within the range of error, and from the slope of T_1ρ_ vs. the inverse temperature, the activation energy E_a_ for the rotational motion below 400 K was determined to be E_a_ = 3.11 ± 0.15 kJ/mol.

The previously reported ^1^H T_1ρ_ values for C_2_H_5_ and NH_3_ of (C_2_H_5_NH_3_)_2_*M*Cl_4_ (*M* = Cu, Zn, and Cd) are shown in Figure 5 as a function of the inverse temperature. More specifically, the ^1^H T_1ρ_ values in the presence of the paramagnetic Co^2+^ and Cu^2+^ ions are particularly short, i.e., 0.01–20 ms, while those of the non-paramagnetic Zn^2+^ and Cd^2+^ ions are longer, i.e., 2–200 ms. In addition, the ^1^H T_1ρ_ values for C_2_H_5_ are longer than those for NH_3_. In contrast, the relaxation times for the ^1^H nuclei in the presence of *M* = Cu, Zn, and Cd reach minimum values, unlike in the case of Co^2+^. For (C_2_H_5_NH_3_)_2_CuCl_4_, the T_1ρ_ for the ^1^H nucleus reaches its minimum values at 190 and 200 K for C_2_H_5_ and NH_3_, respectively, while for (C_2_H_5_NH_3_)_2_ZnCl_4_, the minimum values of 2.17 and 2.48 ms were reached at 260 and 330 K, respectively. Moreover, in case of (C_2_H_5_NH_3_)_2_CdCl_4_, the T_1ρ_ shows a minimum value at 270 K. It is therefore apparent that the ^1^H T_1ρ_ values for (*M* = Cu, Zn, and Cd) vary due to molecular motion according to the Bloembergen–Purcell–Pound (BPP) theory [30], while no such molecular motion is observed for the (*M* = Co) species. Indeed, the T_1ρ_ values are related to the corresponding values of the rotational correlation time, τ_C_, which is a direct measure of the rate of molecular motion. The experimental value of T_1ρ_ can therefore be expressed in terms of τ_C_ for the molecular motion as suggested by the BPP theory [26,29,31,32,33].
T_1ρ_^−1^ = F{4*f*(ω_1_) + *f*(ω_H_ − ω_C_) + 3*f*(ω_C_) + 6*f*(ω_H_ + ω_C_) + 6*f*(ω_H_)}(6)
*f*(ω_1_) = τ_C_/(1 + ω_1_^2^τ_C_^2^),*f*(ω_H_ − ω_C_) = τ_C_/[1 + (ω_H_ − ω_C_)^2^τ_C_^2^],*f*(ω_C_) = τ_C_/(1 + ω_C_^2^τ_C_^2^),*f*(ω_H_ + ω_C_) = τ_C_/[1 + (ω_H_ + ω_C_)^2^τ_C_^2^],*f*(ω_H_) = τ_C_/(1 + ω_H_^2^τ_C_^2^).
where the quantities *f*(ω) are spectral density functions, i.e., Fourier transforms of the time correlation functions. ω_H_ and ω_C_ are the Larmor frequencies of proton and carbon, respectively, and ω_1_ is the frequency of the spin-locking field. The parameter τ_C_ is a characteristic correlation time, that is, the time scale of the motion of the C_2_H_5_ and NH_3_ ions. F is defined as a relaxation constant:
F = (N/20)(γ_H_ γ_C_ħ*/r_H-_*_C_^3^)^2^(7)
where γ_H_ and γ_C_ are the proton and carbon gyromagnetic ratios, respectively, N is the number of directly bound protons, *r*_H__–C_ is the H–C internuclear distance, and ħ is the reduced Planck constant. The obtained data were analyzed assuming that T_1ρ_ has a minimum at ω_1_τ_C_ = 1, and the BPP relationship was applied between T_1ρ_ and the characteristic frequency ω_1_. The value of the relaxation constant F was therefore obtained using Equation (7). From these results, the temperature dependences of the τ_C_ values for the rotational motions of C_2_H_5_ and NH_3_ were calculated from the F values. The temperature dependence of τ_C_ follows the simple Arrhenius equation:
τ_C_ = τ_0_ exp(E_a_/RT)(8)
where E_a_ is the activation energy, τ_0_ is the high temperature limit of the correlation time, T is the temperature, and R is the gas constant. The slope of the linear portion of the semi-log plot represents the E_a_, and the E_a_ for the rotational motion can be obtained from the log τ_C_ vs. 1000/T curve. Thus, the calculated E_a_ values for the four compounds are summarized in Table 3; the activation energies for molecular motion in the presence of paramagnetic Co^2+^ and Cu^2+^ ions were smaller than those for the species containing Zn^2+^ and Cd^2+^.

### 2.3. Investigation of the Structural Properties and Molecular Dynamics by ^13^C CP/MAS NMR

The structural analysis of (C_2_H_5_NH_3_)_2_CoCl_4_ was also performed using ^13^C CP/MAS NMR over a range of increasing temperatures. Thus, the two peaks corresponding to the CH_3_ and CH_2_ species at 360 K were observed at chemical shifts of δ = 49.65 and 176.55 ppm, respectively, as shown in the inset of Figure 6. The CH_3_ and CH_2_ results obtained by ^13^C MAS NMR were distinguished in that the signals corresponding to CH_2_ could not be observed at low temperatures. In these experiments, the chemical shift of CH_3_ remained relatively constant, while that of CH_2_ decreased with increasing temperature, and a sharp decrease was observed close to 420 K.

To obtain the corresponding ^13^C T_1ρ_ values, the nuclear magnetization recovery traces were measured as a function of the delay time. The signal intensities of the magnetization recovery curves for ^13^C were analyzed by a single exponential function of Equation (5) at all temperatures, and the ^13^C T_1ρ_ values for CH_3_ and CH_2_ in (C_2_H_5_NH_3_)_2_CoCl_4_ were plotted as a function of inverse temperature (see Figure 7). Indeed, the ^13^C T_1ρ_ curve for CH_3_ at low temperatures can be reproduced by the BPP theory [32], and the BPP curve shows a minimum of 0.57 ms at 260 K. This characteristic of T_1ρ_ means that distinct molecular motions existed. The correlation time was then obtained using Equation (6), and the activation energy was obtained from these results. More specifically, the E_a_ for the rotational motion was determined to be 45.98 ± 1.78 kJ/mol from the log τ_C_ vs. 1000/T curve shown in Figure 7.

The T_1ρ_ values of the previously reported (C_2_H_5_NH_3_)_2_*M*Cl_4_ (*M* = Cu, Zn, and Cd) (see Figure 8) were compared with those of (C_2_H_5_NH_3_)_2_CoCl_4_ determined herein. In addition, the molecular motions influenced by ^13^C T_1ρ_ in (C_2_H_5_NH_3_)_2_CoCl_4_ were found to exhibit BPP trends, unlike in the case of the ^1^H T_1ρ_ results. Furthermore, for (C_2_H_5_NH_3_)_2_CuCl_4_, the temperature dependences of the ^13^C T_1ρ_ values for CH_2_ and CH_3_ appeared similar, and the BPP curves for CH_3_ and CH_2_ showed minima at 190 K. The T_1ρ_ curve for (C_2_H_5_NH_3_)_2_ZnCl_4_ can be also represented by the BPP theory, with a minimum being observed at 260 K in the curve. Finally, in case of (C_2_H_5_NH_3_)_2_CdCl_4_, the T_1ρ_ curves show minima at 260 and 250 K for CH_3_ and CH_2_, respectively. The ^13^C T_1ρ_ and E_a_ values obtained from the ^13^C results for the four compounds are summarized in Table 2, whereby it is apparent that the ^13^C T_1ρ_ values for compounds containing paramagnetic ions are shorter than those without paramagnetic ions, since the relaxation time should be inversely proportional to the square of the magnetic moment of the paramagnetic ions. Therefore, the T_1ρ_ values of (C_2_H_5_NH_3_)_2_*M*Cl_4_ (*M* = Co and Cu) were driven by fluctuations of the magnetic dipoles of the paramagnetic Co^2+^ and Cu^2+^ species, and the E_a_ values for ^13^C decreased upon increasing the mass of the *M*^2+^ ion, unlike in the case of the ^1^H E_a_ values. These differences are due to variations in the electronic structures of the *M*^2+^ ions, and in particular, the *d* electrons, which screen the nuclear charge from the motion of the outer electrons.

## 3. Materials and Methods

Single crystals of (C_2_H_5_NH_3_)_2_*M*Cl_4_ (*M* = Co, Cu, Zn, and Cd) were grown from CH_3_CH_2_NH_2_∙HCl (ethylamine hydrochloride, Aldrich 98%), and CoCl_2_ (cobalt chloride, Aldrich 97%), CuCl_2_ (copper chloride, Aldrich 97%), ZnCl_2_ (zinc chloride, Aldrich 98%), and CdCl_2_ (cadmium chloride, Aldrich 99.99%), respectively, which were weighed in stoichiometric proportions at 300 K. These crystals were obtained by slow evaporating aqueous solutions containing of CH_3_CH_2_NH_2_ HCl and *M*Cl_2_ in the molar ratio of 2:1.

The thermodynamic properties were measured by TGA (TA, Q600) and optical polarizing microscopy. The differential scanning calorimetry (DSC) and TGA data were recorded between 300 and 770 K under a N_2_ atmosphere using a heating rate of 10 °C/min.

The ^1^H MAS NMR and ^13^C CP/MAS NMR spectra for the rotating frame of (C_2_H_5_NH_3_)_2_*M*Cl_4_ were measured at the Larmor frequencies of 400.13 and 100.61 MHz, respectively, using a Bruker 400 MHz Avance II+ NMR spectrometer (BRUKER, Germany) at the Korea Basic Science Institute, Western Seoul Center. The powder samples were placed in a 4 mm MAS probe, and the MAS rate was set at 10 kHz for the ^1^H MAS and ^13^C CP MAS measurements to minimize any overlap of the spinning sidebands with respect to the central peak. The chemical shifts are listed using tetramethylsilane (TMS) as an internal reference. The spin-lattice relaxation times T_1ρ_ for the rotating frame of (C_2_H_5_NH_3_)_2_*M*Cl_4_ were determined using a π/2−*t* sequence by variation of the spin-locking pulses. The NMR spectra and T_1ρ_ values were recorded between 180 and 430 K.

## 4. Conclusions

We herein discussed the thermodynamic, physical, and structural properties of (C_2_H_5_NH_3_)_2_*M*Cl_4_ (*M* = Co, Cu, Zn, and Cd) layered hybrid materials, where we replaced Pb with nontoxic *M* metals for the production of lead-free perovskite solar cells, and investigated their potential toward solar cell applications based on NMR studies.

The temperature of T_d_ and the degree of mass loss for the decomposition of the 2HCl moieties were both found to depend on the *M* ion present in the structure. Furthermore, the cation dynamics in layered (C_2_H_5_NH_3_)_2_*M*Cl_4_ single crystals were investigated as a function of temperature by ^1^H MAS NMR and ^13^C CP/MAS NMR experiments. To obtain detailed information regarding the cation dynamics of these crystals, the T_1ρ_ values for both ^1^H and ^13^C were obtained, revealing that these atoms undergo rotational motion.

The reason why ^1^H T_1ρ_ of C_2_H_5_ is longer than ^1^H T_1ρ_ of NH_3_ is as follows; the rotational motion for C_2_H_5_ is activated, and that for NH_3_ at the end of the organic cation is less strongly activated. In addition, the reason why ^13^C T_1ρ_ of CH_2_ is longer than ^13^C T_1ρ_ of CH_3_ is as follows; the amplitude of the cation motion is enhanced at its CH_3_ end, and the central CH_2_ moiety is fixed to the NH_3_ group in the organic cation.

Overall, it was found that all components of this series exhibit an orthorhombic structure at room temperature. However, the lattice constants of the crystals containing Co^2+^ and Zn^2+^ ions differed from those of the crystals containing Cu^2+^ and Cd^2+^ ions. It was also found that the inorganic frames of the *M* = Cu^2+^ and Cd^2+^ species are corner-sharing *M*Cl_6_ octahedra, while those of *M* = Co^2+^ and Zn^2+^ are simple *M*Cl_4_ tetrahedra. Finally, it was concluded that the physical properties of these species depend on the characteristics of the organic cation and the inorganic metal ion, but are independent of the arrangements of the *M*Cl_4_ tetrahedra and the *M*Cl_6_ octahedra. The presence of different paramagnetic ions and different lattice constants may also account for these differences.

## Figures and Tables

**Figure 1 molecules-25-01812-f001:**
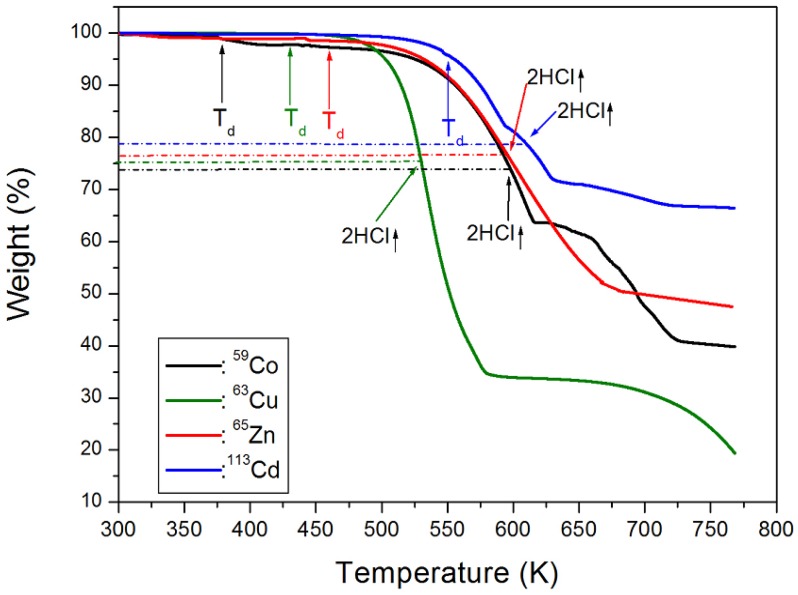
Thermogravimetric analysis (TGA) curve for crystals of (C_2_H_5_NH_3_)_2_*M*Cl_4_ (*M* = Co, Cu, Zn, and Cd).

**Figure 2 molecules-25-01812-f002:**
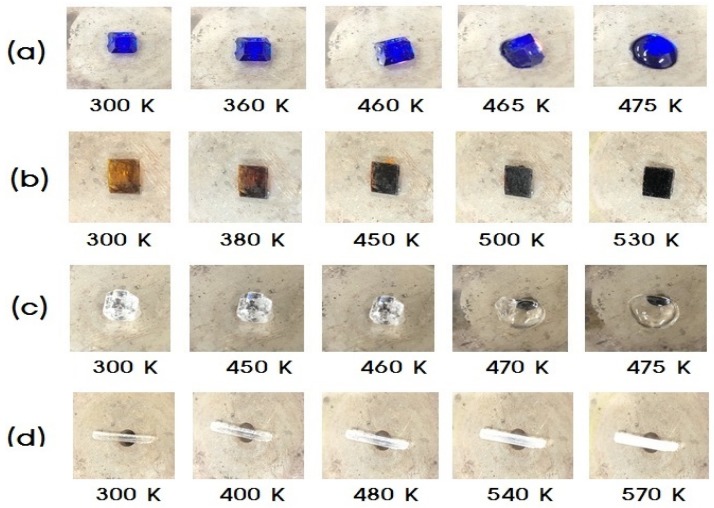
The states of single crystals according to the temperature (**a**) (C_2_H_5_NH_3_)_2_CoCl_4_, (**b**) (C_2_H_5_NH_3_)_2_CuCl_4_, (**c**) (C_2_H_5_NH_3_)_2_ZnCl_4_, (**d**) (C_2_H_5_NH_3_)_2_CdCl_4_.

**Figure 3 molecules-25-01812-f003:**
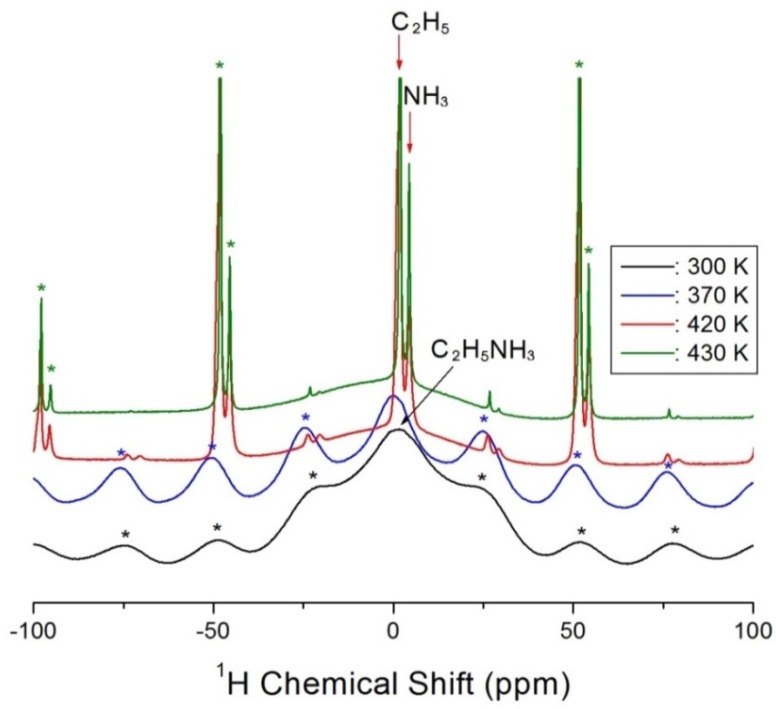
^1^H magic angle spinning (^1^H MAS) NMR spectra of (C_2_H_5_NH_3_)_2_CoCl_4_ at 300 K, 370 K, 420 K, and 430 K. The spinning sidebands for central peak are marked with asterisk.

**Figure 4 molecules-25-01812-f004:**
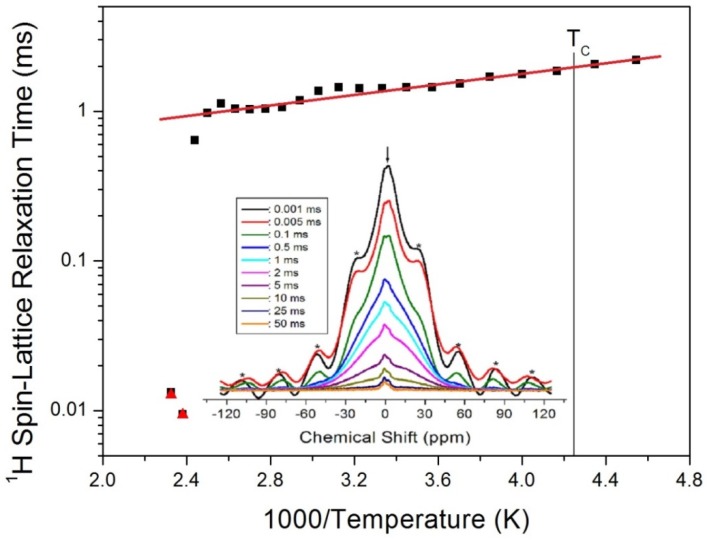
^1^H spin-lattice relaxation times T_1ρ_ in the rotating frame in C_2_H_5_NH_3_ cation of (C_2_H_5_NH_3_)_2_CoCl_4_ as a function of inverse temperature. The black square and red triangle at 410 K and 420 K is for ^1^H T_1ρ_ in the C_2_H_5_ and NH_3_ group, respectively (inset: the ^1^H recovery traces according to the delay times at 300 K).

**Figure 5 molecules-25-01812-f005:**
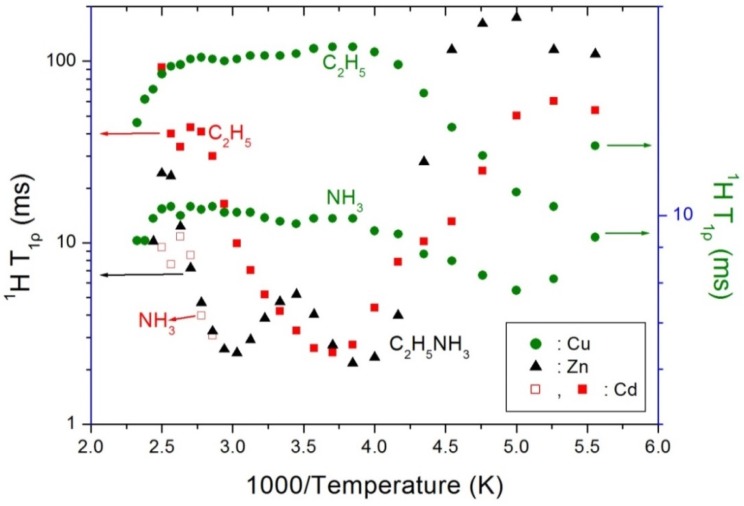
^1^H spin-lattice relaxation times T_1ρ_ in the rotating frame in (C_2_H_5_NH_3_)_2_*M*Cl_4_ (*M* = Cu, Zn, and Cd) as a function of inverse temperature.

**Figure 6 molecules-25-01812-f006:**
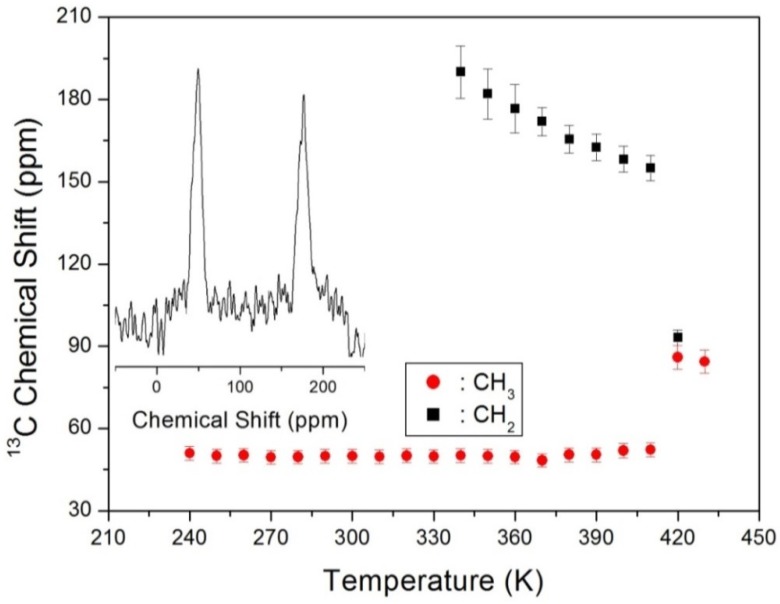
^13^C chemical shift in CH_3_ and CH_2_ groups in (C_2_H_5_NH_3_)_2_CoCl_4_ as a function of temperature (inset: ^13^C MAS NMR spectrum at 360 K).

**Figure 7 molecules-25-01812-f007:**
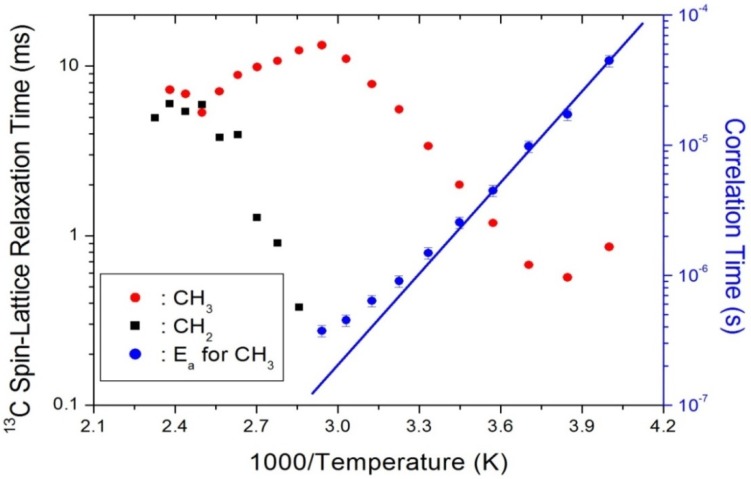
^13^C spin-lattice relaxation times in the rotating frame for CH_3_ and CH_2_ groups in (C_2_H_5_NH_3_)_2_CoCl_4_ as a function of inverse temperature (inset: Arrhenius plots of the natural logarithm of the correlation time for CH_3_ as a function of inverse temperature).

**Figure 8 molecules-25-01812-f008:**
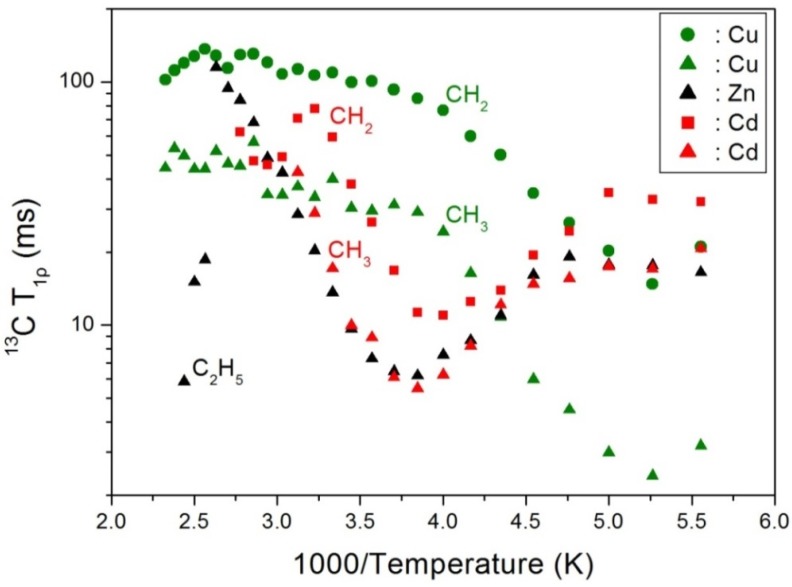
^13^C spin-lattice relaxation times T_1ρ_ in the rotating frame in (C_2_H_5_NH_3_)_2_*M*Cl_4_ (*M* = Cu, Zn, and Cd) as a function of inverse temperature.

**Table 1 molecules-25-01812-t001:** The phase transition temperatures, T_C_, lattice constants, structures, and space groups of (C_2_H_5_NH_3_)_2_*M*Cl_4_ (*M* = Co, Cu, Zn, and Cd) crystals at room temperature.

*M*	T_C_ (K)	Lattice Constant (Å)			Structure	Space Group
Co	235	a = 10.025	b = 7.403	c = 17.603	orthorhombic	*Pnma*
Cu	236, 330, 357, 371	a = 7.47	b = 7.35	c = 21.18	orthorhombic	*Pbca*
Zn	231, 234, 237, 247, 312	a = 10.043	b = 7.397	c = 17.594	orthorhombic	*Pna2_1_*
Cd	114, 216, 358, 470	a = 7.354	b = 7.478	c = 22.11	orthorhombic	*Abma*

**Table 2 molecules-25-01812-t002:** The decomposition temperature, T_d_, mass loss of 2HCl, and final decomposition product *M*Cl_2_ for four crystals.

*M*	T_d_ (K)	Weight Loss of 2HCl (%) (cal. Value)	Weight Loss of 2HCl (%) (exp. Value)	Final Decomposition Product (%) (cal. Value)
Co	378	25.97	26.09	53.76
Cu	430	24.51	24.84	54.81
Zn	460	24.36	23.17	54.47
Cd	550	21.05	21.00	47.08

**Table 3 molecules-25-01812-t003:** The spin-lattice relaxation times, T_1ρ_, and activation energies, E_a_, for ^1^H and ^13^C of (C_2_H_5_NH_3_)_2_*M*Cl_4_ (*M* = Co, Cu, Zn, and Cd) crystals.

*M*	^1^H T_1ρ_ (ms)	E_a_ (kJ/mol)	^13^C T_1ρ_ (ms)	E_a_ (kJ/mol)
Co	0.01–2	3.11 (for C_2_H_5_NH_3_)	0.1–10	45.98 (for CH_3_)
Cu	7–20	12.19 (for C_2_H_5_ below 240 K)	1–100	21.35 (for CH_3_)
		8.33 (for NH_3_ below 240 K)		19.72 (for CH_2_)
Zn	2–200	39.41 (for C_2_H_5_NH_3_ above 290 K)	6–100	21.13 (for C_2_H_5_)
		57.59 (for C_2_H_5_NH_3_ below 290 K)		
Cd	2–100	22.63 (for C_2_H_5_NH_3_)	5–100	18.05 (for CH_3_)

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
