# Peer review of "Thermodynamic, Physical, and Structural Characteristics in Layered Hybrid Type (C2H5NH3)2MCl4 (M = 59Co, 63Cu, 65Zn, and 113Cd) Crystals"

_molecules, 2020, doi:10.3390/molecules25081812_

Round 1

Reviewer 1 Report

Interesting work presenting physicochemical characteristics of new compounds  due to their potential application in solar cells.

Author Response

Reviewer 1

There is no reviewer’s comment.

Reviewer 2 Report

The manuscript entitled "Thermodynamic, physical, and structural  characteristics in layered hybrid type (C2H5NH3) 2MCl4  (M = 59Co, 63Cu, 65Zn, and 113Cd) crystals" reports a study  on four different of layered hybrid  types  that differ in the nature of the inorganic bivalent cation.

This samples were investigated by thermogravimetric analysis (TGA) and magic angle spinning nuclear magnetic resonance (MAS NMR) spectroscopy.

The particularly detailed work could be interesting for researchers working in this sector given the amount of data and information that is provided.

I believe, however, that there are some small gaps that the author will have to perfect and therefore I  say that it can be considered for publication but only after a minor revision.

Here are some tips:

- Introduction: there is a repetition of data reported in the manuscript (line 49-59) and table 1. I suggest to report only the data in table 1 with the respective bibliographical notes.

- Are equations 1-4 supported by literature references?

-Lines 116-126. This part should be improved. I suggest inserting a table with weight loss and peaks for the different samples sorted by size of the cation. It could also be useful to insert a graph where you can appreciate the linear variations of the thermal parameters as a function of the size of the cation.

The values shown ( lines 119)  are very close to each other (e.g. 24.51%, 24.36%). In order to be sure that they actually represent different trends, it is necessary to specify how many times the thermal analyzes have been repeated and what the error is.

Line 121. The text reports "The final decomposition product is MCl2, which corresponds to mass losses of 53.76, 54.81, 54.47, and 47.08%"  but to whom are these values related?

Line 122. The text reports "These results indicate some differences between the calculated and  experimental values", how can this difference be justified?

Figure 1.I find it unclear due to the overlap of the curves. The thermal curve of the zinc (red line) appears incomplete. In this case how was the final loss calculated?

Figure 1 shows the weight% or weight loss% in the ordinate?

Figure 2. How were these samples treated? How long have they been maintained at the set temperature?

Materials and methods. More details on sample preparation must be provided.

Author Response

Reviewer 2

The manuscript entitled "Thermodynamic, physical, and structural characteristics in layered hybrid type (C2H5NH3) 2MCl4 (M=59Co, 63Cu, 65Zn, and 113Cd) crystals" reports a study on four different of layered hybrid types that differ in the nature of the inorganic bivalent cation.

This samples were investigated by thermogravimetric analysis (TGA) and magic angle spinning nuclear magnetic resonance (MAS NMR) spectroscopy.

The particularly detailed work could be interesting for researchers working in this sector given the amount of data and information that is provided.

I believe, however, that there are some small gaps that the author will have to perfect and therefore I say that it can be considered for publication but only after a minor revision.

Here are some tips:

Q1: Introduction: there is a repetition of data reported in the manuscript (line 49-59) and table 1. I suggest to report only the data in table 1 with the respective bibliographical notes.

A1: line 49-59 “As an example, (C2H5NH3)2CoCl4 crystallizes as an orthorhombic structure with the Pnma space group, which undergoes a reversible phase transition at 235 K; the lattice constants are a=10.0256 Å, b=7.4030 Å, and c=17.6039 Å at room temperature [14]. In addition, (C2H5NH3)2CuCl4 undergoes phase transitions at 236, 330, 357, and 371 K [7, 15–19], its crystal structure at room temperature is orthorhombic with the Pbca space group, and the lattice constants are a=7.47 Å, b=7.35 Å, and c=21.18 Å [20]. In contrast, (C2H5NH3)2ZnCl4 undergoes five phase transitions at 231, 234, 237, 247, and 312 K [21], crystallizing as an orthorhombic system with the Pna21 space group at room temperature, and with lattice parameters of a=10.043 Å, b=7.397 Å, c=17.594 Å, and Z=4 [22]. Finally, (C2H5NH3)2CdCl4 undergoes structural phase transitions at 114, 216, 358, and 470 K [9, 23, 24], whereby the room temperature orthorhombic phase has the Abma space group with lattice constants of a=7.354 Å, b=7.478 Å, and c=22.11 Å [23].” is revised to “As an example, (C2H5NH3)2CoCl4 crystallizes as an orthorhombic structure, which undergoes a reversible phase transition at 235 K [14]. In addition, (C2H5NH3)2CuCl4 undergoes phase transitions at 236, 330, 357, and 371 K [7, 15–19], its crystal structure at room temperature is orthorhombic [20]. In contrast, (C2H5NH3)2ZnCl4 undergoes five phase transitions at 231, 234, 237, 247, and 312 K [21], crystallizing as an orthorhombic system at room temperature [22]. Finally, (C2H5NH3)2CdCl4 undergoes structural phase transitions at 114, 216, 358, and 470 K [9, 23, 24], whereby the room temperature orthorhombic phase has the Abma space group [23].”

Q2: Are equations 1-4 supported by literature references?

A2: Reference for equations 1-4 is added as [28].

Q3: Lines 116-126. This part should be improved. I suggest inserting a table with weight loss and peaks for the different samples sorted by size of the cation. It could also be useful to insert a graph where you can appreciate the linear variations of the thermal parameters as a function of the size of the cation.

The values shown ( lines 119)  are very close to each other (e.g. 24.51%, 24.36%). In order to be sure that they actually represent different trends, it is necessary to specify how many times the thermal analyzes have been repeated and what the error is.

A3-1: Table 2 is added. And, Td is inserted in Fig. 1.

A3-2: For the M=Co, Cu, Zn, and Cd species, the first mass losses were observed at approximately 378, 430, 460, and 550 K, respectively, which represent the onset of partial thermal decomposition, Td. It was found that the temperature of Td increases with the divalent metal M ions. From the results calculated using the molecular weights, mass losses of 25.97, 24.51, 24.36, and 21.05% for the different M ions are shown by dotted lines in Fig. 1, and these were attributed to decomposition of the 2HCl moieties. The final decomposition product is MCl2, which corresponds to mass losses of 53.76, 54.81, 54.47, and 47.08%. These results indicate some differences between the calculated and experimental values. ” is revised to “For the M=Co, Cu, Zn, and Cd species, the first mass losses were observed at approximately 378, 430, 460, and 550 K, respectively, which represent the onset of partial thermal decomposition, Td. From the results calculated using the molecular weights, mass losses of 25.97, 24.51, 24.36, and 21.05 % for the different M ions were attributed to decomposition of the 2HCl moieties. These results are consistent with the TGA experiment results shown by dotted lines in Fig. 1. And, the final decomposition product is MCl2, which corresponds to mass losses of 53.76, 54.81, 54.47, and 47.08 %. These results indicate some differences between the calculated and experimental values. The difference between the calculation and experimental value of the final decomposition product is presumably dependent on the heating rate in the TGA experiment. Another difference is thought to be due to experimental conditions in air or N2 atmosphere. The decomposition temperature, Td, and mass loss of 2HCl, and final decomposition product for four crystals are summarized in Table 2.”

Q4: Line 121. The text reports "The final decomposition product is MCl2, which corresponds to mass losses of 53.76, 54.81, 54.47, and 47.08%"  but to whom are these values related?

A4: These values are calculated using Eqs. (1)-(4).

Q5: Line 122. The text reports "These results indicate some differences between the calculated and  experimental values", how can this difference be justified?

A5: The difference between the calculation and experimental value of the final decomposition product is presumably dependent on the heating rate in the TGA experiment. Another difference is thought to be due to experimental conditions in air or N2 atmosphere.

Q6: Figure 1.I find it unclear due to the overlap of the curves. The thermal curve of the zinc (red line) appears incomplete. In this case how was the final loss calculated?

A6: The TGA curve for Zn is added completed. Final loss is calculated by the molecular weights in Eqs. (1)-(4).

Q7: Figure 1 shows the weight% or weight loss% in the ordinate?

A7: “Weight Loss (%)” in Fig. 1 is revised to “Weight (%).

Q8: Figure 2. How were these samples treated? How long have they been maintained at the set temperature?

A8: “Here, the sample temperatures shown in Fig. 2 were kept constant during 2 min each temperature.” is added in Page 4.

Q9: Materials and methods. More details on sample preparation must be provided.

A9: “These crystals were obtained by slow evaporating aqueous solutions containing of CH3CH2NH2·HCl and MCl2 in the molar ratio of 2:1.” is added in Page 9.

Reviewer 3 Report

Comments to the Authors

In the papaer  “Thermodynamic, physical, and structural characteristics in layered hybrid type (C2H5NH3)2MCl4 (M= 59Co, 63Cu, 65Zn, and 113Cd) crystals " The thermal, physical, and molecular dynamics of layered hybrid type (C2H5NH3)2MCl4  (M=Co, Cu, Zn, and Cd)  crystals  were  investigated  by  thermogravimetric  analysis  (TGA)  and  magic angle spinning nuclear magnetic resonance (MAS NMR) spectroscopy.

The manuscript addresses an interesting topic and describes clearly the experimental methods and results. Therefore, it may be recommended for publication after minor revision:

Pag1, Line 11: there aren’t the numbers that indicate the isotopes (for example the author wrote Co instead of 59Co)

Pag1, Lines 40-41: The metals divalent should be written as Co2+, Cu2+, Zn2+ and Cd2+

Pag2, Line 64: Write M in italic

Pag2, Line 71: The c value should be written with three numbers after the dot

Pag2, Line 80: (CP)/MAS should be changed with (CP/MAS)

Pag2, Line 90: change was calculated with were calculate because of the plural subject

Pag4, Line 125: Write M in italic. Adjust the subscripts of the molecular formula

Pag4, Lines 142-143: Add the commas among the four different crystals

Pag4, Line 152: Was should be were because of the plural subject

Pag5, Line 163: Change “are plotted” into “were plotted” because the tense is not respected

Pag5, Line 165: Change “were” into “are” because the tense is not respected

Pag 6: Fig.4 is not very clear. The colors should be more discernible, so adjust the resolution. In addition the legend of the curves is not enjoyable to read

Pag7, Line 206: Write M in italic

Pag8, Line 231: Write M in italic

Pag8, Line 243: 13C is not correct. You should write 13C

Pag10, Line 279: Write M in italic. Adjust the subscript of the molecular formula

Pag10, Line 292: Delete the dash between 4 and mm

Pag10, Line 300: Write M in italic

Pag10, Line 305: Substitute “More specifically” with a synonym

Pag11, Line 341:in reference 2. You forgot to write 2+ as apex (Cu2+)

Pag11, Line344: There is a mistake in the word “different” and the author forgot to write all the subscripts into the molecular formula

Pag11, Line 347: in the reference 5.there is a mistake in the word “Ferrolectric”. It should be written “Ferroelectric”

Pag11, Line 359: in reference 10. correct the word “optical”

Pag12, Line 375: in reference 16, the author forgot to write the year of the work in bold type

Pag12, Line 381: in reference 19.delete the [ and put a space

Pag12, Line 388: in reference 23.  space between the words Electrical and properties is necessarily

Author Response

Reviewer 3

In the papaer  “Thermodynamic, physical, and structural characteristics in layered hybrid type (C2H5NH3)2MCl4 (M= 59Co, 63Cu, 65Zn, and 113Cd) crystals " The thermal, physical, and molecular dynamics of layered hybrid type (C2H5NH3)2MCl4 (M=Co, Cu, Zn, and Cd) crystals were investigated by thermogravimetric  analysis  (TGA)  and  magic angle spinning nuclear magnetic resonance (MAS NMR) spectroscopy.

The manuscript addresses an interesting topic and describes clearly the experimental methods and results. Therefore, it may be recommended for publication after minor revision:

Q1: Pag1, Line 11: there aren’t the numbers that indicate the isotopes (for example the author wrote Co instead of 59Co)

A1: I have indicated isotope according to the reviewer’s comment.

Q2: Pag1, Lines 40-41: The metals divalent should be written as Co2+, Cu2+, Zn2+ and Cd2+

A2: Modified according to the reviewer’s comment.

Q3: Pag2, Line 64: Write M in italic

A3: Modified according to the reviewer’s comment.

Q4: Pag2, Line 71: The c value should be written with three numbers after the dot

A4: Modified according to the reviewer’s comment.

Q5: Pag2, Line 80: (CP)/MAS should be changed with (CP/MAS)

A5: Modified according to the reviewer’s comment.

Q6: Pag2, Line 90: change was calculated with were calculate because of the plural subject

A6: Modified according to the reviewer’s comment.

Q7: Pag4, Line 125: Write M in italic. Adjust the subscripts of the molecular formula

A7: Modified according to the reviewer’s comment.

Q8: Pag4, Lines 142-143: Add the commas among the four different crystals

A8: Modified according to the reviewer’s comment.

Q9: Pag4, Line 152: Was should be were because of the plural subject

A9: Modified according to the reviewer’s comment.

Q10: Pag5, Line 163: Change “are plotted” into “were plotted” because the tense is not respected

A10: Modified according to the reviewer’s comment.

Q11: Pag5, Line 165: Change “were” into “are” because the tense is not respected

A11: Modified according to the reviewer’s comment.

Q12: Pag 6: Fig.4 is not very clear. The colors should be more discernible, so adjust the resolution. In addition the legend of the curves is not enjoyable to read

A12: Figure 4 has been modified to be clear.

Q13: Pag7, Line 206: Write M in italic

A13: Modified according to the reviewer’s comment.

Q14: Pag8, Line 231: Write M in italic

A14: Modified according to the reviewer’s comment.

Q15: Pag8, Line 243: 13C is not correct. You should write 13C

A15: Modified according to the reviewer’s comment.

Q16: Pag10, Line 279: Write M in italic. Adjust the subscript of the molecular formula

A16: Modified according to the reviewer’s comment.

Q17: Pag10, Line 292: Delete the dash between 4 and mm

A17: Modified according to the reviewer’s comment.

Q18: Pag10, Line 300: Write M in italic

A18: Modified according to the reviewer’s comment.

Q19: Pag10, Line 305: Substitute “More specifically” with a synonym

A19-1: “More specifically, we examined the crystal growth of these (C2H5NH3)2MCl4 species in addition to comparing their TGA and NMR properties according to the metal ions present. “ is deleted.

A19-2: “More specifically, the thermodynamic properties were investigated through the use of TGA, and the temperature of Td and the degree of mass loss for the decomposition of the 2HCl moieties were both found to depend on the M ion present in the structure.” is revised to “The temperature of Td and the degree of mass loss for the decomposition of the 2HCl moieties were both found to depend on the M ion present in the structure.”

Q20: Pag11, Line 341:in reference 2. You forgot to write 2+ as apex (Cu2+)

A20: Modified according to the reviewer’s comment.

Q21: Pag11, Line344: There is a mistake in the word “different” and the author forgot to write all the subscripts into the molecular formula

A21: Modified according to the reviewer’s comment.

Q22: Pag11, Line 347: in the reference 5.there is a mistake in the word “Ferrolectric”. It A22:

should be written “Ferroelectric”

A22: Modified according to the reviewer’s comment.

Q23: Pag11, Line 359: in reference 10. correct the word “optical”

A23: Modified according to the reviewer’s comment.

Q24: Pag12, Line 375: in reference 16, the author forgot to write the year of the work in bold type

A24: Modified according to the reviewer’s comment.

Q25: Pag12, Line 381: in reference 19.delete the [ and put a space

A25: Modified according to the reviewer’s comment.

Q26: Pag12, Line 388: in reference 23.  space between the words Electrical and properties is necessarily

A26: Modified according to the reviewer’s comment.

Reviewer 4 Report

This paper shows an interesting topic related to the structural characteristic of layered hybrid. Studies of RMN are well presented and discussed. However it lacks of a deeper study related to the thermodynamic and physic characteristic.

TGA and Optical Polarizing microscopy techniques

1. The presented analyzes of the TGA and Optical Polarizing microscopy techniques are only observations, a deeper analysis is required. How are these analyzes related to the rest of the results presented in the work?

2. Figure 1 requires better and greater analysis, clearly indicating the different stages or thermal events (the temperatures and weight losses observed in the thermograms), from the first weight loss, to the final residue and assigning them to the different events, if possible use references that support this.
Indicate the temperature of thermal decomposition of the material (elimination of organics).

i.e. indicate the intersection between temperature and loss of mass (to clarify the sentence on line 117),
Indicate in the figure the weight loss and the temperature corresponding to the removal of 2HCl (the thermogram does not indicate an inflection point. You can put the derivative to clarify this...........

2.Why is the calculated mass loss not consistent with that observed in Figure 1?

3. Line 118......”It was found that the temperature of Td increases with the divalent metal M ions.”.....

this is not understood, should be clarified by the authors

4. Line140,.....” decomposition is one of the key chemical reactions taking place on the crystal surface.

decomposition is only carried on the surface of the crystals?

5. In which temperature are the crystals thermally stable?

Conclusions

6. The conclusion are very long. I mean, they are very long and should be more concrete and conclusive. They are presented as observations and summary of the results obtained from the different techniques used

7.Lines 305-307 .......”More specifically, the thermodynamic properties were investigated through the use of TGA, 306 and the temperature of Td and the degree of mass loss for the decomposition of the 2HCl moieties 307 were both found to depend on the M ion present in the structure.

The use of the TGA technique and the analysis presented in this work based on this technique does not represent a study of the thermodynamic properties of crystals.

Author Response

Reviewer 4

This paper shows an interesting topic related to the structural characteristic of layered hybrid. Studies of RMN are well presented and discussed. However it lacks of a deeper study related to the thermodynamic and physic characteristic.

TGA and Optical Polarizing microscopy techniques

Q1: The presented analyzes of the TGA and Optical Polarizing microscopy techniques are only observations, a deeper analysis is required. How are these analyzes related to the rest of the results presented in the work?

A1-1: For the M=Co, Cu, Zn, and Cd species, the first mass losses were observed at approximately 378, 430, 460, and 550 K, respectively, which represent the onset of partial thermal decomposition, Td. It was found that the temperature of Td increases with the divalent metal M ions. From the results calculated using the molecular weights, mass losses of 25.97, 24.51, 24.36, and 21.05% for the different M ions are shown by dotted lines in Fig. 1, and these were attributed to decomposition of the 2HCl moieties. The final decomposition product is MCl2, which corresponds to mass losses of 53.76, 54.81, 54.47, and 47.08%. These results indicate some differences between the calculated and experimental values. ” is revised to “For the M=Co, Cu, Zn, and Cd species, the first mass losses were observed at approximately 378, 430, 460, and 550 K, respectively, which represent the onset of partial thermal decomposition, Td. From the results calculated using the molecular weights, mass losses of 25.97, 24.51, 24.36, and 21.05 % for the different M ions were attributed to decomposition of the 2HCl moieties. These results are consistent with the TGA experiment results shown by dotted lines in Fig. 1. And, the final decomposition product is MCl2, which corresponds to mass losses of 53.76, 54.81, 54.47, and 47.08 %. These results indicate some differences between the calculated and experimental values. The difference between the calculation and experimental value of the final decomposition product is presumably dependent on the heating rate in the TGA experiment. Another difference is thought to be due to experimental conditions in air or N2 atmosphere. The decomposition temperature, Td, and mass loss of 2HCl, and final decomposition product for four crystals are summarized in Table 2.”

A1-2: Table 2 is added.

A1-3: Optical polarizing microscopy was then conducted to further understand the thermal stabilities of the various species at high temperatures, as presented in Fig. 2.” is revised to “Optical polarizing microscopy was used in order to determine whether these transformations are structural phase transitions or chemical reactions, as presented in Fig. 2.”

A1-4: “For all four crystals, it was apparent that the phenomenon above Td was not related to any structural phase transitions, but rather to a thermal decomposition, whereby the solid-state decomposition is one of the key chemical reactions taking place on the crystal surface.” is revised to “For all four crystals, it was apparent that the phenomenon above Td was not related to any structural phase transitions, but rather to a thermal decomposition, suggested by Lee [29].”

A1-5: Reference [29] is added.

Q2: Figure 1 requires better and greater analysis, clearly indicating the different stages or thermal events (the temperatures and weight losses observed in the thermograms), from the first weight loss, to the final residue and assigning them to the different events, if possible use references that support this.
Indicate the temperature of thermal decomposition of the material (elimination of organics).

i.e. indicate the intersection between temperature and loss of mass (to clarify the sentence on line 117),
Indicate in the figure the weight loss and the temperature corresponding to the removal of 2HCl (the thermogram does not indicate an inflection point. You can put the derivative to clarify this...........

Why is the calculated mass loss not consistent with that observed in Figure 1?

A2-1: The Td is added in Figure 1.

A2-2: Same as the answer to Q1.

A2-3: Table 2 is added.

Q3: Line 118......”It was found that the temperature of Td increases with the divalent metal M ions.”.....

this is not understood, should be clarified by the authors

A3: This result was inadvertently increased with M ions, and it has no physical meaning, so that “It was found that the temperature of Td increases with the divalent metal M ions.” is deleted.

Q4: Line140,.....” decomposition is one of the key chemical reactions taking place on the crystal surface.

decomposition is only carried on the surface of the crystals?

A4: The decomposition does not only act on the surface of the crystal. To reduce this misunderstanding, “For all four crystals, it was apparent that the phenomenon above Td was not related to any structural phase transitions, but rather to a thermal decomposition, whereby the solid-state decomposition is one of the key chemical reactions taking place on the crystal surface.” is revised to “For all four crystals, it was apparent that the phenomenon above Td was not related to any structural phase transitions, but rather to a thermal decomposition, suggested by Lee [29].”

Q5: In which temperature are the crystals thermally stable?

A6: I think it will be thermally stable below Td.

Conclusions

Q6: The conclusion are very long. I mean, they are very long and should be more concrete and conclusive. They are presented as observations and summary of the results obtained from the different techniques used

A: The conclusion was made concrete and conclusive.

A6-1: “More specifically, we examined the crystal growth of these (C2H5NH3)2MCl4 species in addition to comparing their TGA and NMR properties according to the metal ions present.” is deleted.

A6-2: “More specifically, the thermodynamic properties were investigated through the use of TGA, and” is deleted.

A6-3: “The BPP curves for the 1H T of C2H5 and NH3 and for the 13C T of CH3 and CH2 were shown to exhibit minima as a function of the inverse temperature. This implies that these curves represent the rotational motions of 1H and 13C.” is deleted.

A6-4: “Furthermore, the T values and activation energies obtained from the 1H measurements for the H‒Cl···M (M=Zn and Cd) bond in (C2H5NH3)2MCl4 in the absence of paramagnetic ions were larger than those for the paramagnetic M=Co and Cu. Moreover, the Ea value for 13C, which is distant from the M2+ ions, was found to decrease upon increasing the mass of the M2+ ion, unlike in the case of the Ea values for 1H.” is deleted.

A6-5: “These differences are due to variations in the electronic structures of the M2+ ions, and in particular, the d electrons, which screen the nuclear charge from the motion of the outer electrons.” is moved to “Result and Discussion” in Page 9

Q7:.Lines 305-307 .......”More specifically, the thermodynamic properties were investigated through the use of TGA, 306 and the temperature of Td and the degree of mass loss for the decomposition of the 2HCl moieties 307 were both found to depend on the M ion present in the structure.

The use of the TGA technique and the analysis presented in this work based on this technique does not represent a study of the thermodynamic properties of crystals

A7-1: “More specifically, we examined the crystal growth of these (C2H5NH3)2MCl4 species in addition to comparing their TGA and NMR properties according to the metal ions present.” is deleted.

A7-2: “More specifically, the thermodynamic properties were investigated through the use of TGA, and the temperature of Td and the degree of mass loss for the decomposition of the 2HCl moieties were both found to depend on the M ion present in the structure.” is revised to “The temperature of Td and the degree of mass loss for the decomposition of the 2HCl moieties were both found to depend on the M ion present in the structure.”

Round 2

Reviewer 4 Report

The present revised manuscript has improved, the authors followed the suggestions and made the pertinent corrections.

The manuscript can be accepted in the present form.